# Beyond Parameter Count:
# Implicit Bias in Soft Mixture of Experts

## Abstract

The traditional viewpoint on Sparse Mixture of Experts (MoE) models is that instead of training a single *large* expert, which is computationally expensive, we can train many *small* experts. The hope is that if the total parameter count of the small experts equals that of the singular large expert, then we retain the representation power of the large expert while gaining computational tractability and promoting expert specialization. The recently introduced Soft MoE replaces the Sparse MoE's discrete routing mechanism with a differentiable gating function that smoothly mixes tokens. While this smooth gating function successfully mitigates the various training instabilities associated with Sparse MoE, it is unclear whether it induces implicit biases that affect Soft MoE's representation power or potential for expert specialization. We prove that Soft MoE with a single arbitrarily powerful expert cannot represent simple convex functions. This justifies that Soft MoE's success cannot be explained by the traditional viewpoint of many small experts collectively mimicking the representation power of a single large expert, and that multiple experts are actually *necessary* to achieve good representation power (even for a fixed total parameter count). Continuing along this line of investigation, we introduce a notion of expert specialization for Soft MoE, and while varying the number of experts yet fixing the total parameter count, we consider the following (computationally intractable) task. Given any input, how can we discover the expert subset that is specialized to predict this input's label? We empirically show that when there are many small experts, the architecture is implicitly biased in a fashion that allows us to efficiently approximate the specialized expert subset. Our method can be easily implemented to potentially reduce computation during inference. For example, using our method on ImageNet, one can perform inference using only $1/8$ of the experts and still retain 99% of the test accuracy of using all experts.

## 1    Introduction

It has been well established that scaling the size (i.e., parameter count) of models is necessary for state of the art prediction power (Kaplan et al., 2020; Bahri et al., 2021), but naively scaling the model size is infeasible due to hardware constraints and computational costs. Mixture of Experts (MoE) layers in a model allow one to achieve this goal, while mitigating the increased computational costs for training and inference that accompany a naively larger model. These have been successfully deployed in practice, in a variety of contexts such as language (Fedus et al., 2022) and vision (Ruiz et al., 2021), and MoE layers are considered critical to achieve state of the art performance in today's models (Jiang et al., 2024).

The archetypical MoE layer is the Sparse MoE (Shazeer et al., 2017). Here, each token is only given to a subset of experts, and a router is trained to discretely match a token to its expert subset. Since a single (large) expert can represent complex functions, the traditional viewpoint is that one should partition the large expert into multiple (small) experts, so that the total parameter count of all experts is unchanged. The hope is that the model's representation power is similar since the total parameter count is the same and since experts can specialize to the tokens they see (rather than all tokens) (Chen et al., 2022). Yet, training and inference are faster, because any single token activates only a subset of experts rather than all of them.

While the Sparse MoE allows scaling of model size, its discrete routing causes optimization issues and load balancing difficulties during training. To tackle these issues, many variants of Sparse MoE have been introduced, such as routing a token to only a single expert (Fedus et al., 2022), incorporating linear programs to ensure load balancing (Lewis et al., 2021) or having the experts select tokens (instead of tokens selecting experts) (Zhou et al., 2022). However, all these approaches remain discrete in nature, and thus suffer from at least some degree of training instability.

To alleviate these issues, the recently introduced Soft MoE (Puigcerver et al., 2024) eschews discrete matching in favor of a smoother approach. It computes for each expert a convex combination of the input tokens, and the expert only sees this convex combination. The final output of the model is then a convex combination of each expert's output. This approach is fully differentiable, and hence is more stable than the Sparse MoE. This novel Soft MoE architecture has been shown to outperform all other baselines on challenging large scale vision tasks, and can scale to thousands of experts (Puigcerver et al., 2024). Moreover, recent results show that the Soft MoE is a promising avenue towards providing empirical scaling laws for deep reinforcement learning (Obando-Ceron et al., 2024).

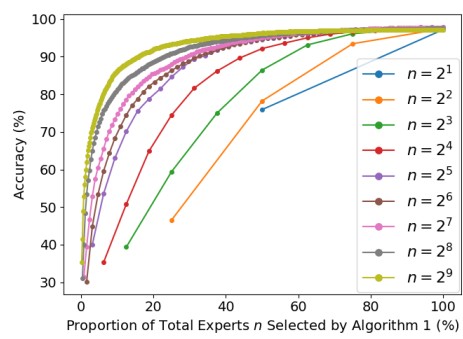

Figure 1: Our Algorithm 1 selects a *specialized* subset of the experts to utilize for inference. Given any proportion of $n$ (the total number of experts) to select, its performance uniformly improves with larger $n$.

Thus, while the Sparse MoE constructs a discrete mapping between tokens and experts, the Soft MoE computes convex combinations of tokens that are fed to experts, and then computes convex combinations of the expert outputs, which together promote stabler and faster training.

The majority of prior work on MoE focuses on computational issues, such as efficient and stable training. In our paper, we adopt an orthogonal perspective. In particular, it remains unclear whether Soft MoE's specific manner of combining tokens and experts creates any unexpected implicit architectural biases. Indeed, it is not even clear that its soft gating mechanism preserves the traditional MoE dogma that many small experts have similar representation power to a single large expert with the same total parameter count. It is also unclear whether combining tokens and experts completely destroys the possibility (or discoverability) of expert specialization, which is what one traditionally desires from an MoE (especially in the regime of many experts) (Krishnamurthy et al., 2023; Dai et al., 2024). Thus, we investigate for the existence of such biases, through the lens of *varying the number of experts*. In this paper, we make progress along this line of investigation by making the following contributions:

- We prove that the Soft MoE with a single neural network expert, even with arbitrarily many parameters, cannot represent simple convex functions (while empirically we show that multiple experts can). Thus, in contrast to the traditional viewpoint, having multiple experts is actually necessary to have non-trivial representation power in Soft MoE.

- We introduce a notion of specialization for Soft MoE. While discovering specialized experts generally seems intractable, we empirically demonstrate that as we increase the number of experts, even while fixing the total parameter count, the architecture is implicitly biased in a manner that allows us to efficiently approximate the specialized expert set (see Figure 1).

- Our method for discovering specialized experts can be easily implemented for reducing computation at inference.

These contributions thus show there are benefits to using a large number of small experts relative to a small number of large experts, and notably, these benefits are often non-computational.

## 2 PROBLEM FORMULATION

### 2.1 PRELIMINARIES

We begin by briefly discussing the Soft MoE architecture (Puigcerver et al., 2024). Throughout our paper, we assume there is a single slot per expert, since this is the most performant setting in practice. Let $X \in \mathbb{R}^{m \times d}$ denote the tokenized input, so that there are $m$ tokens each in $\mathbb{R}^d$. The MoE layer is equipped with $n$ experts $\{f_j : \mathbb{R}^d \to \mathbb{R}^d\}_{j=1}^n$, each of which is typically implemented as a feedforward network. The router is parameterized by $\Phi \in \mathbb{R}^{d \times n}$. Given an input $X$, the parameters $\Phi$ are used to compute matrices $D(X), C(X) \in \mathbb{R}^{m \times n}$ which are defined elementwise as

$$D(X)_{ij} = \frac{\exp\left((X\Phi)_{ij}\right)}{\sum_{i'=1}^m \exp\left((X\Phi)_{i'j}\right)} \quad \text{and} \quad C(X)_{ij} = \frac{\exp\left((X\Phi)_{ij}\right)}{\sum_{j'=1}^n \exp\left((X\Phi)_{ij'}\right)}. \tag{1}$$

Note that each column of $D(X)$ and each row of $C(X)$ sums to one. With this notation in hand, we formally define the Soft MoE layer below.

**Definition 1** *The Soft MoE is a function* $\text{sMoE}_{\{f_j\}_{j=1}^n}^{\Phi} : \mathbb{R}^{m \times d} \to \mathbb{R}^{m \times d}$ *defined as*

$$\text{sMoE}_{\{f_j\}_{j=1}^n}^{\Phi}(X) = C(X)\widetilde{Y}(X) \quad \text{where} \quad \widetilde{Y}(X) = \begin{bmatrix} f_1\left((D(X)^T X)_1\right) \\ \vdots \\ f_n\left((D(X)^T X)_n\right) \end{bmatrix}.$$

The Soft MoE thus computes $n$ different convex combinations of the tokens in $X$, where the weights of the $j$th convex combination are given by the $j$th column of $D(X)$. It then applies expert $f_j$ to the $j$th convex combination, for each $j = 1, 2 \ldots n$. Finally, it computes $m$ different convex combinations of these expert outputs, where the weights of the $i$th convex combination are given by the $i$th row of $C(X)$. Note that each expert processes a single vector in $\mathbb{R}^d$, and that sMoE is differentiable whenever the experts are. This results in more stable training relative to Sparse MoE, where each expert is given a subset of the $m$ tokens via a discrete matching algorithm. The Soft MoE has shown significant empirical success in vision (Puigcerver et al., 2024) and reinforcement learning (Obando-Ceron et al., 2024).

### 2.2 OUR INVESTIGATION

At a high level, the Sparse MoE is designed with the following principle. Say we desire a model with $b$ total parameters, because we believe that $b$ allows for sufficiently large representation power. Instead of using a single large network ($n = 1$) with $b$ parameters, we use $n > 1$ smaller experts each with $b/n$ parameters. The hope is that we have similar representation power to the $n = 1$ case since the total parameter count is the same, but we have faster computation because each token only activates a small subset of experts (Shazeer et al., 2017; Fedus et al., 2022). Moreover, one also hopes that each expert can specialize to the specific type of tokens it sees (Chen et al., 2022; Krishnamurthy et al., 2023; Dai et al., 2024).

The Soft MoE is motivated in a similar fashion. It also splits a single large model with $b$ parameters into $n \geq 1$ smaller experts each with $b/n$ parameters, hoping that representation power is unchanged. However, Soft MoE differs significantly from Sparse MoE in how it uses these experts. While Sparse MoE discretely assigns tokens to experts, Soft MoE computes convex combinations of tokens and expert outputs. Due to this significant difference, it is unclear how the original motivations for Sparse MoE apply to Soft MoE.

As one example of how Soft MoE might deviate from the original motivations for Sparse MoE, consider the extreme case when $n = 1$. Here, Sparse MoE's router trivially routes all tokens to the expert, and so classical results show that Sparse MoE can represent arbitrary continuous functions (Cybenko, 1989; Hornik, 1991). But in Soft MoE, the gating function is non-trivial even in $n = 1$, and so it is possible that Soft MoE has poor representation power even when the expert is very powerful. If this were true, then it would challenge the conventional wisdom that Soft MoE's empirical success with $n > 1$ smaller experts (each with $b/n$ parameters) is simply because they

mimic (albeit computationally efficiently) the representation power of a single large expert (with $b$ parameters). Instead, it would suggest that Soft MoE has some implicit bias that enables its practical success. To this end, we ask the following question.

> *Q1: Can a large single expert in Soft MoE represent simple functions?*

Our motivation for this question has thus far primarily been scientific. Nevertheless, we believe this question is also practically relevant, since there are empirical results in reinforcement learning (RL) which show that Soft MoE with a single expert can outperform traditional neural network architectures (Obando-Ceron et al., 2024). Indeed, a negative answer to Q1 would imply that there are implicit biases in Soft MoE that assist in its improved empirical performance.

As a second example of how Soft MoE might deviate from the original motivations for Sparse MoE, consider the following. Since Soft MoE combines all tokens before feeding them to an expert, and since it combines all experts to produce its final output, it is natural to wonder whether this prohibits expert specialization. Indeed, this limitation is acknowledged in the seminal work (Puigcerver et al., 2024). We thus ask the following question.

*Q2: Is there a notion of expert specialization in Soft MoE, and if so, can it be efficiently discovered?*

The remainder of this paper is devoted to answering Q1 (in Section 3) and Q2 (in Section 4).

## 3 REPRESENTATION FAILURE OF A SINGLE EXPERT

In this section, we answer Q1 from Section 2.2. We first recall the definition of Lipschitz functions.

**Definition 2** *A function $h : \mathbb{R}^{k_1} \to \mathbb{R}^{k_2}$ is L-Lipschitz if $\|h(x) - h(y)\|_2 \leq L\|x - y\|_2$ for all $x, y \in \mathbb{R}^{k_1}$.*

Recall that any neural network is Lipschitz (Virmaux & Scaman, 2018). Our main result shows that Soft MoE with a single Lipschitz expert $f$ is incapable of representing simple target functions. This result holds even when this expert $f$ is arbitrarily powerful (and possibly non-parametric). It also holds when the output of the Soft MoE layer is passed to an arbitrarily powerful Lipschitz function $g$ (as would be the case in a practical implementation, since the MoE would be a layer that prepends a powerful neural network function, rather than a standalone layer). Below we formally state our result.

**Theorem 1** *Fix any $m \geq 2$, $d \geq 1$ and $n = 1$. Define the target function $t : \mathbb{R}^{m \times d} \to \mathbb{R}$ as $t(X) = \|X\|_2$. Assume the existence of $\Phi \in \mathbb{R}^{d \times 1}$, $f : \mathbb{R}^d \to \mathbb{R}^d$ and $g : \mathbb{R}^{m \times d} \to \mathbb{R}$ such that*

$$\left| t(X) - g\left(\mathrm{sMoE}_f^{\Phi}(X)\right) \right| \leq \max\left\{1, t(X)/20\right\} \text{ for all } X \in \mathbb{R}^{m \times d}.$$

*Then there are no $L_f, L_g \geq 0$ such that $f$ is $L_f$-Lipschitz and $g$ is $L_g$-Lipschitz.*

The proof is deferred to Appendix A. Let us discuss this result.

**Notion of Approximation.** The theorem says that we cannot approximate $t$ over $X \in \mathbb{R}^{m \times d}$, up to an error that scales as $\max\{1, t(X)/20\}$. While the domain is unbounded, which is slightly non-standard, our approximation error is also unbounded since we allow it to scale with $t(X)$ (unlike traditional results which require approximation to within a fixed constant tolerance). Indeed, it is trivial that the constant function zero can approximate $t(X)$ over all of $\mathbb{R}^d$ up to an error of $t(X)$. Our notion of error is only slightly smaller than this.

**Residual Connections.** In a practical implementation, one would use residual connections so that the function $g$ would typically receive both $X$ and $\mathrm{sMoE}_f^{\Phi}(X)$ as input instead of just $\mathrm{sMoE}_f^{\Phi}(X)$. In such a setting, our result would of course not apply, since one could use $g$ alone to approximate $t(X)$, while completely ignoring $\mathrm{sMoE}_f^{\Phi}(X)$. Nevertheless, we believe it is worth studying the setting without residual connections, because if $g$ did not leverage $\mathrm{sMoE}_f^{\Phi}(X)$, then there would be no point of using a Soft MoE layer at all.

**Benign Target.** The target function $t$ is extremely benign. Indeed, it is convex and 1-Lipschitz. So it a priori seems intuitive to try and approximate it with a Lipschitz expert $f$ and subsequent Lipschitz

architecture $g$. Yet, we cannot approximate $t$ even when $f, g$ have arbitrarily large Lipschitz constants. This is in stark contrast to the classical neural network approximation literature, where relatively simple feedforward networks can represent arbitrary continuous functions (Cybenko, 1989; Hornik, 1991).

**Permutation Invariant Target.** Let $\sigma : \mathbb{R}^{m \times d} \to \mathbb{R}^{m \times d}$ be any function that permutes the rows of its input. It is easily shown that $\sigma$ commutes with $\text{sMoE}^{\Phi}_{\{f_j\}_{j=1}^n}$. This implies that one cannot represent target functions that vary based on permutations of their input's rows. A result which relied on this property would not be very satisfying, since in practice this issue is handled by adding positional encoding to the input (Vaswani et al., 2017). However, our target function $t$ satisfies $t(\sigma(X)) = t(X)$ for any permutation function $\sigma$. Indeed, an examination of the proof in Appendix A shows that the result hinges on the particular manner in which tokens are combined before being passed to the expert.

**Normalizing Input.** In practice, one may normalize the input $X$ before passing it to the Soft MoE, and in this case Theorem 1 is trivially inapplicable. Nevertheless, since the performance with or without normalization is typically very comparable (Puigcerver et al., 2024), we believe the result remains interesting.

**Multiple Experts.** We are unable to prove a theorem for the ability of multiple experts to represent $t$. In Appendix B, however, we show empirically that increasing the number of experts leads to monotonic performance improvement for representing $t$, even when the total expert parameter count is fixed.

Theorem 1 (and the experiment in Appendix B) thus provides a negative answer to Q1 raised in Section 2.2. It therefore challenges the conventional wisdom that Soft MoE's empirical success with $n > 1$ smaller experts (each with $b/n$ parameters) is simply because they mimic the representation power of a single large expert (with $b$ parameters). Instead, it suggests that Soft MoE has implicit biases that assist in its improved performance. A precise characterization of these biases and their role in this improvement is beyond our paper's scope. Nevertheless, we view our result as a stepping stone towards understanding and potentially furthering the true power of Soft MoE. Indeed, recent results show that Soft MoE with even a single expert can outperform traditional architectures (Obando-Ceron et al., 2024). Our Theorem 1 shows that this improvement *cannot be because of improved representation power*.

## 4 SPECIALIZATION OF EXPERTS

In this section, we tackle Q2 from Section 2.2.

### 4.1 WHAT DOES SPECIALIZATION MEAN?

We begin by noting that since the Soft MoE combines tokens before feeding them to experts, it seems a priori unlikely that experts specialize, as acknowledged in the seminal work (Puigcerver et al., 2024). Indeed, it is unclear what it even means for an expert to specialize; for instance it seems difficult to consider specialization of an expert to a subset of tokens.

Instead, our notion of specialization is whether there exists, for any input $X \in \mathbb{R}^{m \times d}$, an $X$-dependent (relatively small) subset of experts that are sufficient to accurately predict the label for that input $X$. Recall from Definition 1 that the output of the Soft MoE layer is $C(X)\widetilde{Y}(X)$, where $\widetilde{Y}(X) \in \mathbb{R}^{n \times d}$ stores the output of expert $i$ in row $i$. Hence, zeroing out a row $i$ of $\widetilde{Y}(X)$ means that expert $i$ does not contribute anything to the final prediction. If we can zero out many rows of $\widetilde{Y}(X)$ without affecting the final prediction, then the remaining experts can be understood to have specialized to the input $X$, since this means that only these remaining experts were actually required to make an accurate prediction, and these experts did not require contributions from the other zeroed out experts. By contrast, if zeroing out rows of $\widetilde{Y}(X)$ changes the prediction, then this indicates a lack of expert specialization, since the knowledge required to predict correctly on $X$ was non-trivially spread out across each of the $n$ experts.

## 4.2 DOES SPECIALIZATION EXIST?

To test whether specialization occurs, we consider the following small experiment. For $n \in \{4, 8, 16\}$ we train a simple neural network that comprises of a Soft MoE layer with $n$ experts, followed by a linear prediction head, on the MNIST dataset (LeCun et al., 2010). The experts in the MoE are each (identical architecture) MLP with one hidden layer and ReLU activation, and we denote the number of parameters in a single expert to be $d_{E,n}$. As we increase $n$, we decrease the width of each expert (i.e. the number of units in the hidden layer), so that the total (i.e., summed over all experts) number of parameters $nd_{E,n}$ is constant for all values of $n$. By holding the total number of expert parameters constant, we keep the total expressivity constant, and thus ensure that we do not bias the results for larger numbers of experts. See Appendix D.1 for further details of this experimental setup.

We train each network, and for the sake of a fair comparison across $n$, we select for each $n$ a model that has $\approx 97.5\%$ test accuracy. We emphasize that our aim is to investigate the impact of $n$ on expert specialization, thus we need to first ensure that each setting of $n$ achieves the same test performance. Then, for each of the $n$ models and each test datapoint $X$ we do the following: (a) we use an exhaustive search to identify whether there exists an expert subset of size $k = n/4$ that predicts the label for $X$ correctly (b) we randomly zero out $3n/4$ rows of $\widetilde{Y}(X)$, so we are only predicting using a random set of $k = n/4$ experts, and check whether it predicts the label for $X$ correctly. For both settings, we compute the average prediction accuracy over all $10,000$ test points. Note for both settings that the number of expert parameters involved in prediction is invariant to the number of experts $n$.

The first row of Table 1 shows that expert specialization occurs, especially for larger $n$ (even though each of the experts for larger $n$ are smaller in parameter count). Figure 5 in Appendix D.1 shows that the best $k$-subsets identified are diverse, indicating the expert subsets are indeed specialized to the input. However, the exhaustive search used to generate this result is generally intractable for larger models. It is also not useful for prediction in the wild, since this procedure requires knowledge of the label of each test point $X$. Moreover, the second row of Table 1 shows that we cannot hope to find the best subset simply via random selection.

| | $n = 4$ | $n = 8$ | $n = 16$ |
|---|---|---|---|
| Best $k$-Subset Accuracy (%) | 94.69 | 99.90 | 100.00 |
| Random $k$-Subset Accuracy (%) | $46.57 \pm 0.44$ | $51.95 \pm 0.46$ | $58.94 \pm 0.34$ |

Table 1: Results for experiment in Section 4.2, where a subset of the experts of size $k = n/4$ was used to predict the labels. For the Random subset results, we report the mean and standard deviation over 10 random seeds.

## 4.3 AN ALGORITHM FOR BEST EXPERT SELECTION

The results of the experiment in Section 4.2 demonstrate the existence of expert specialization, albeit in a small experiment. However, the same results show that identifying the best subset of experts cannot be done as simply as via random selection. Since identifying the true best subset of $k$ experts ostensibly has $\Omega\left(\binom{n}{k}\right)$ computational complexity, it is of interest to discover an efficient algorithm which can rapidly approximate the best subset, especially before moving on to larger experiments. Beyond our current motivations of checking for the existence of expert specialization, such an algorithm could also be useful at inference time to reduce computational expense without loss in accuracy (see Section 4.5).

To this end, we recall from Definition 1 that given an input $X$, the final output of the Soft MoE is $C(X)\widetilde{Y}(X)$, i.e. row $i$ of the final output is a convex combination of the rows of $\widetilde{Y}(X)$ where the combination weights are given by row $i$ of $C(X)$. So a natural attempt to identify the most important experts are those given the most weight by $C(X)$. Algorithm 1 formalizes this intuition.

Note that Algorithm 1 is computationally efficient, requires no separate training, and can be implemented in just a few lines of code. We also note that it can be easily adapted to handle a batched input in a vectorized fashion (see Appendix C). Its output $\widehat{Y}(X) \in \mathbb{R}^{n \times d}$ equals $\widetilde{Y}(X)$ on $k$ rows, and is zero in the other $n - k$ rows. Thus, Algorithm 1 can be used to identify a performant subset of experts at inference time, and then construct the final Soft MoE output by only doing a forward

---

**Algorithm 1** Best Expert Subset Selection

---

**Require:** number of experts $k$ to use for prediction, Soft MoE $\text{sMoE}^{\Phi}_{\{f_j\}_{j=1}^n}$, input $X \in \mathbb{R}^{m \times d}$

1: Compute $C(X) \in \mathbb{R}^{m \times n}$ as in Equation 1.
2: Define $C_{\text{sum}} \in \mathbb{R}^n$ entrywise as $C_{\text{sum},j} = \sum_{i=1}^m C(X)_{ij}$ for $j = 1, 2 \ldots n$.
3: Define $S_k \subseteq \{1, 2 \ldots n\}$ to be the indices corresponding to the $k$ largest entries of $C_{\text{sum}}$.
4: Define $\widehat{Y}(X) \in \mathbb{R}^{n \times d}$ entrywise as

$$\widehat{Y}(X)_{ij} = \begin{cases} \left( f_i \left( (D(X)^T X)_i \right) \right)_j & \text{if } i \in S_k \\ 0 & \text{if } i \notin S_k \end{cases},$$

for each $i = 1, 2 \ldots n$ and $j = 1, 2 \ldots d$.
5: **return** $\widehat{Y}(X)$.

---

pass through $k$ experts instead of all $n$ experts. In the regime of many large experts, this can yield computational advantages at inference time, and we explore this further in Section 4.5.

## 4.4 EMPIRICAL PERFORMANCE OF ALGORITHM 1

In line with the experimental setup assumed by the seminal work (Puigcerver et al., 2024), we demonstrate the efficacy of Algorithm 1 on a suite of image classification tasks. We provide results across a wide range of model scales and architectures, including architectures beyond the ViT model class that was used to introduce Soft MoE.

### 4.4.1 EXPERIMENTAL SETUP

We experiment on 4 datasets: MNIST, CIFAR10, CIFAR100 (Krizhevsky, 2009) and ImageNet-1k (Deng et al., 2009). For MNIST, we use the same experimental setup as in Section 4.2. Recall that the network used is very simple, and consists entirely of a Soft MoE layer and a prediction head. This small network has merely 318K total parameters, of which 307K are expert parameters. As a more practical setting, we use the Astroformer-1 architecture (Dagli, 2023) for CIFAR10 and CIFAR100. This hybrid transformer-convolutional architecture achieves excellent performance on these datasets without leveraging extra training data. We modify Astroformer-1 by replacing the MLPs in the transformer blocks with Soft MoE layers, analogous to the standard practice that is followed in ViTs (Puigcerver et al., 2024). This model is larger, and has 180M total parameters, of which 150M are expert parameters. Finally, we adopt the same Soft MoE variant of the ViT architecture (Dosovitskiy et al., 2021) used by the seminal work (Puigcerver et al., 2024) on the ImageNet-1k dataset. Specifically, it replaces the MLPs in the latter half of the encoder blocks of the ViT Base model with Soft MoE layers. This is the largest architecture we consider, and it has 513M total parameters, of which 454M are expert parameters. Due to compute constraints, scaling up our experimental protocol further (either with external pretraining data or larger model sizes) is infeasible. Nevertheless, our setup spans a range of model architecture types and scales, as well as several canonical datasets. We defer further details of the various architectures and their modifications to Appendix D.2.

For each dataset and associated network architecture, we do the following. We train a suite of models where each model has a different $n$ (total number of experts) in each Soft MoE layer. Each model is trained *from scratch* without utilizing any external data. As discussed in Section 4.2, when we increase $n$ we correspondingly decrease the number of parameters in each expert, to ensure the total expert expressivity is (roughly) constant. After training each network, we select for each $n$ a model that has the same test accuracy (as discussed in Section 4.2, this is important because we are investigating the impact of varying $n$ on Algorithm 1's test performance, and so we must ensure the original networks have similar test performance). For each of these trained networks and various values of $k$, we then evaluate Algorithm 1's performance on the test set. Concretely, we compute the test accuracy of using the $k$ experts found by Algorithm 1 for each datapoint in the test set, and compare this to the test accuracy of randomly selecting $k$ experts for each datapoint in the test set.

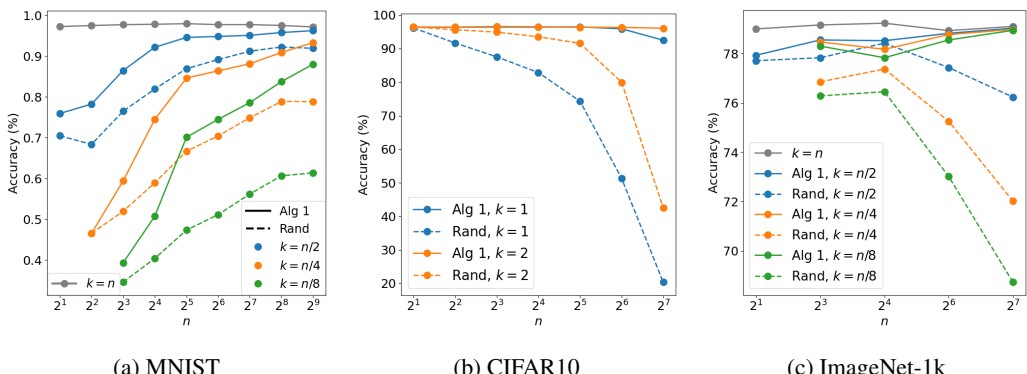



(a) MNIST      (b) CIFAR10      (c) ImageNet-1k



Figure 2: Results for MNIST, CIFAR10 and ImageNet-1k experiments in Section 4.4. We depict the test accuracy as a function of $n$, for Algorithm 1 and Random selection, and for various choices of $k$. For the Random selection results, we report the mean over 10 random seeds. For CIFAR10, we only reported results with $k = 1$ and $k = 2$. This is because $k > 2$ had accuracies that were nearly identical to using all experts. We hypothesize this is because CIFAR10 is relatively easy for the powerful Astroformer architecture.

### 4.4.2 EXPERIMENTAL RESULTS

In Figures 2 and 3 we depict, for each dataset and various choices of $k$, the performance of Algorithm 1 relative to the random selection scheme. We make two main observations.

The first observation is that Algorithm 1's accuracy may deteriorate (relative to using all experts) when $k < n$ and $n$ is small. But if $n$ is big, then Algorithm 1's accuracy is often very close to that of using all the experts, even when $k \ll n$. For instance, on MNIST, using the $k = n/2$ experts selected by Algorithm 1 gives nearly the same performance as using all experts when $n \geq 64$, but has poor performance when $n \leq 8$. This result consistently holds across the various datasets/architectures/model scales considered: e.g. Table 13b shows that on ImageNet-1k, when $n$ is high enough, Algorithm 1 can use just $1/8$ of the experts while retaining over $99\%$ of the original accuracy of using all the experts. This shows that as $n$ increases, Algorithm 1 is better poised to discover specialized experts, even though the number of expert parameters is invariant to $n$.

We believe this is an instance of the implicit bias that exists in Soft MoE. Concretely, the experts identified by Algorithm 1 for an input $X$ are those highest weighted by $C_{\text{sum}}$, which is derived from $C(X)$. Our result thus shows that as $n$ increases, the Soft MoE trains in a manner such that the $C(X)$ matrix is more informative for which experts are useful in correctly predicting the label of $X$, even though such a property is not explicitly enforced during training! In Appendix D.4, we show that the skewness of $C_{\text{sum}}$ does not change significantly as $n$ increases, which suggests that the specialization identified by Algorithm 1 is not trivial by simply dropping marginally weighted experts. While we do not have a formal explanation for why this phenomenon occurs, we conjecture that this is an implicit bias that allows the specialized subsets to be more easily identified by $C_{\text{sum}}$ as $n$ increases. A different view of the same results in Figure 2a is provided in Figure 1.

The second observation is that Algorithm 1's test performance dominates that of random selection. And often, Algorithm 1 is far better than random selection, as is particularly evident in CIFAR10 and ImageNet-1k, where random selection is very poor (see Figure 2). There are instances where random selection has decent accuracy, such as in CIFAR100 (see Figure 3). So to further validate our result, we use the following statistic to measure how much better Algorithm 1 is relative to random selection. We first compute the standard deviation of the random selection test accuracies (which were obtained by averaging over 10 random seeds). Our statistic is then the number of standard deviations by which Algorithm 1 exceeds the mean random selection accuracy. When this statistic is large, it means that Algorithm 1 found an expert subset whose performance is statistically significantly larger than that of the typical random expert subset. The results are shown in Table 2 where we observe very large values for this statistic. This shows that our Algorithm 1 significantly outperforms random selection for CIFAR100, even though random selection has decent performance on this dataset. Analogous tables for the other datasets are provided in Appendix D.3.

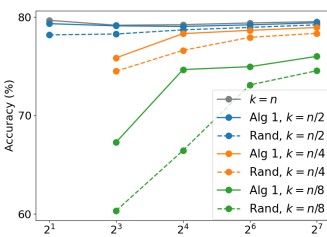

Figure 3: Results for CIFAR100 experiments from Section 4.4.

| | $n$ | | | | |
|---|---|---|---|---|---|
| | 2 | 8 | 16 | 64 | 128 |
| $k = n/2$ | 6.23 | 6.73 | 1.89 | 3.47 | 2.49 |
| $k = n/4$ | - | 7.31 | 6.34 | 4.56 | 4.50 |
| $k = n/8$ | - | 21.22 | 28.20 | 13.96 | 5.74 |

Table 2: Each cell is the number of standard deviations that the test accuracy of Algorithm 1 is above the mean test accuracy of Random selection, on the CIFAR100 dataset. For the Random selection, we used 10 random seeds to obtain its mean and standard deviation.

### 4.5 FASTER INFERENCE VIA ALGORITHM 1

In practice, one typically trains a very large model (with many Soft MoE layers, each with many experts) a single time, then the model is deployed and repeatedly utilized for inference. It is thus of great interest to speed up computation through Soft MoE modules, since this leads to faster inference. Algorithm 1 is well suited for this purpose. The preceding sections show that when the number of experts $n$ is large, then Algorithm 1 can rapidly and cheaply find a small subset of experts that can predict any datapoint with minimal expected accuracy loss. So it can potentially be used for faster inference through the Soft MoE.

A full examination of the usefulness of Algorithm 1 for faster inference is beyond the scope of our paper. A proper analysis would require massive industry-scale models, and would also be very application specific because it would depend on the type and number of devices used for inference, as well as the extent of distributed computing and per-device parallelization. We lack the compute capacity to do such a thorough study. Nevertheless, it is immediate that in the regime of many large experts, where a forward pass through an expert has non-trivial cost, Algorithm 1 should save computation since its overhead is relatively negligible. As a simple example to show the potential of Algorithm 1 for faster inference, we consider the same ViT architecture from Section 4.4 with 8 total experts, but with each expert being much larger, such that the whole model has 2.32B parameters, of which 2.27B parameters are in the experts. We measure the total wall clock time elapsed during a forward pass through the 6 Soft MoE layers, where each layer will only utilize $k$ experts as prescribed by Algorithm 1. The results are presented in Table 3, and show that smaller values of $k$ yield significant speedups. See Appendix D.5 for further details on this experiment.

| | $k = n$ | $k = 3n/4$ | $k = n/2$ | $k = n/4$ |
|---|---|---|---|---|
| batch = 1 | $21.50 \pm 0.20$ | $19.46 \pm 0.23$ | $14.75 \pm 0.35$ | $9.96 \pm 0.19$ |
| batch = 100 | $99.03 \pm 0.28$ | $78.26 \pm 0.51$ | $78.86 \pm 0.36$ | $51.69 \pm 0.79$ |

Table 3: Wall clock time of performing a single forward pass (of random data) through the 6 Soft MoE layers using Algorithm 1, with either a single datapoint or a batch of 100 datapoints. The unit of all values are milliseconds. For each batch size, we report the mean and standard deviation over 100 trials.

## 5 RELATED WORK

**Mixture of Experts.** Our work falls squarely in the literature on MoEs, which originated several decades ago (Jacobs et al., 1991; Jordan & Jacobs, 1993) and has been revived as the Sparse MoE (Shazeer et al., 2017). Nevertheless, there is a significant difference between our investigation and the majority of past work. The majority of past work in MoE focuses primarily on computational considerations, such as (in the context of Sparse MoE) routing a token to only a single expert for efficiency (Fedus et al., 2022), and ensuring load balancing by incorporating linear programs (Lewis et al., 2021) or having the experts select tokens (Zhou et al., 2022) or re-routing dropped tokens (Zeng et al., 2024). Indeed, the Soft MoE (Puigcerver et al., 2024) was recently introduced to address the various training instabilities suffered by Sparse MoE, and performs very well in applications like vision (Puigcerver et al., 2024), RL (Obando-Ceron et al., 2024) and audio processing (Cappellazzo et al., 2024). By contrast, our investigation begins with a more fundamental question – what are the

implicit biases that occur as a result of Soft MoE's particular manner of combining tokens and expert outputs? While analogous fundamental investigations of the gating function do exist in the context of Sparse MoEs (Chen et al., 2022; Dikkala et al., 2023) and some of its variants (Nguyen et al., 2023; 2024), to the best of our knowledge this line of investigation is novel in Soft MoE.

**Expert Specialization.** The use of MoE layers in a network is often motivated by the desire for expert specialization, since this implies a more efficient use of the network parameters. To our knowledge, all prior work on expert specialization has been conducted in the context of Sparse MoE. While some studies demonstrate that expert specialization may occur for Sparse MoE (Chen et al., 2022; Dikkala et al., 2023), others show that specialization is often limited and requires architectural innovations (Krishnamurthy et al., 2023; Dai et al., 2024; Oldfield et al., 2024). These innovations are different than Soft MoE, where it is not even a priori clear what expert specialization means. Indeed, the seminal work acknowledges this limitation (Puigcerver et al., 2024). In our paper, we offer a notion of expert specialization in Soft MoE, and show not only that it occurs but can be efficiently detected.

**Sparse Activation & Pruning.** Our work is also related to the literature on sparsely activating or pruning a network, since our Algorithm 1 can be viewed as a form of pruning at inference. For instance, there is work on using RL to conditionally activate only certain units in a generic network for faster training and inference (Bengio et al., 2016), pruning CNN kernels for efficient inference (Molchanov et al., 2017), and developing adaptive computation modules for transformers (Wójcik et al., 2023). While a complete survey of this vast area is beyond the current scope (Blalock et al., 2020), we emphasize that our form of pruning is entirely specific to Soft MoE, since it relies on the $C(X)$ matrix. It can be used with or without many other types of pruning.

## 6 DISCUSSION

**Limitations.** Our work has a number of limitations. While our Theorem 1 is a negative result for a single expert's representation power, we are unable to prove a corresponding positive result for multiple experts (although in Appendix B we provide empirical evidence for increased representation power as the number of experts increases). A different limitation is that we are unable to provide a thorough analysis of the extent to which Algorithm 1 can reduce computation at inference (due to a lack of compute capacity). We believe both limitations provide exciting directions for future work.

**Conclusion.** In this paper, we studied the Soft Mixture of Experts architecture. We eschewed the traditional viewpoint of scaling expert parameter count in a manner that allows efficient training and inference. Instead, we adopted an orthogonal perspective, and studied implicit biases that arise from Soft MoE's particular manner of combining token and expert outputs. We showed that even with an arbitrarily powerful single expert, Soft MoE cannot represent simple convex functions, thus showing that good representation power is predicated on having multiple experts. We also introduced a notion of expert specialization for Soft MoE, and provided an algorithm that efficiently discovers specialized experts when the number of experts is large. Overall, our analysis (both theoretical and empirical) highlights non-traditional reasons for why using many smaller experts is preferable to using fewer larger experts, even when the total expert parameter count remains fixed.

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

## A    PROOF OF THEOREM 1

Here, we formally prove Theorem 1. Assume for the sake of contradiction that there exist $\Phi \in \mathbb{R}^{d \times 1}$, $f : \mathbb{R}^d \to \mathbb{R}^d$ and $g : \mathbb{R}^{m \times d} \to \mathbb{R}$ such that

$$\left| t(X) - g\left(\text{sMoE}_f^\Phi(X)\right) \right| \leq \max\{1, t(X)/20\} \text{ for all } X \in \mathbb{R}^{m \times d}, \tag{2}$$

where $f$ is $L_f$-Lipschitz and $g$ is $L_g$-Lipschitz. Via the $L_g$-Lipschitz property and via Definition 1, we know for any $A, B \in \mathbb{R}^{m \times d}$ that

$$
\begin{aligned}
\left| g\left(\text{sMoE}_f^\Phi(B)\right) - g\left(\text{sMoE}_f^\Phi(A)\right) \right| &\leq L_g \left| \text{sMoE}_f^\Phi(B) - \text{sMoE}_f^\Phi(A) \right| \\
&= L_g \left\| C(B)\widetilde{Y}(B) - C(A)\widetilde{Y}(A) \right\|_2 \\
&= L_g \left\| \begin{bmatrix} 1 \\ 1 \\ \vdots \\ 1 \end{bmatrix} \left( \widetilde{Y}(B) - \widetilde{Y}(A) \right) \right\|_2 \\
&= L_g \sqrt{m} \left\| \widetilde{Y}(B) - \widetilde{Y}(A) \right\|_2 .
\end{aligned}
$$

Now again using Definition 1 and applying the $L_f$-Lipschitz property, we obtain from the above inequality that

$$
\begin{aligned}
\left| g\left(\text{sMoE}_f^\Phi(B)\right) - g\left(\text{sMoE}_f^\Phi(A)\right) \right| &\leq L_g \sqrt{m} \left\| \widetilde{Y}(B) - \widetilde{Y}(A) \right\|_2 \\
&= L_g \sqrt{m} \left\| f\left(D(B)^T B\right) - f\left(D(A)^T A\right) \right\|_2 \\
&\leq L_g L_f \sqrt{m} \left\| D(B)^T B - D(A)^T A \right\|_2 .
\end{aligned} \tag{3}
$$

We now breakup the remainder of the proof into disjoint and exhaustive cases based on the sign of the first entry of $\Phi$, and in each case we show a contradiction.

We begin with the case where $\Phi_1 > 0$. Define $A \in \mathbb{R}^{m \times d}$ to be the matrix which is all zeros, except $A_{11} = a$ and $A_{21} = -a$ for a value of $a > 0$ that is yet to be specified. Then $A\Phi \in \mathbb{R}^{m \times 1}$ is a vector of all zeros except that its first entry is $a\Phi_1 > 0$ and its second entry is $-a\Phi_1 < 0$. By the definition of the matrix $D$ in Section 2.1, this implies for large $a > 0$ that

$$D(A)_i = \begin{cases} 1 - \Omega\left(m \exp(-a)\right) & \text{if } i = 1 \\ \mathcal{O}\left(\exp(-a)\right) & \text{if } i > 1 \end{cases} .$$

Now define $B \in \mathbb{R}^{m \times d}$ to be the matrix which is all zeros, except $B_{11} = a$ for the aforementioned value of $a > 0$ that is yet to be specified. Then $B\Phi \in \mathbb{R}^{m \times 1}$ is a vector of all zeros except that its first entry is $a\Phi_1 > 0$. By the definition of the matrix $D$ in Section 2.1, this implies for large $a > 0$ that

$$D(B)_i = \begin{cases} 1 - \Omega\left(m \exp(-a)\right) & \text{if } i = 1 \\ \mathcal{O}\left(\exp(-a)\right) & \text{if } i > 1 \end{cases} .$$

Let $A_1, B_1$ denote the first column of $A, B$. Leveraging continuity of the absolute value, we thus obtain that

$$
\begin{aligned}
\lim_{a \to \infty} \left\| D(B)^T B - D(A)^T A \right\|_2 &= \lim_{a \to \infty} \left| D(B)^T B_1 - D(A)^T A_1 \right| \\
&= \left| \lim_{a \to \infty} \left( D(B)^T B_1 - D(A)^T A_1 \right) \right| \\
&= 0.
\end{aligned} \tag{4}
$$

Note via definition of $t$ that $t(A) - t(B) = (\sqrt{2} - 1)a$. Now recalling Eq. equation 2 and Eq. equation 3, we know for large $a > 0$ that

$$
\begin{aligned}
(\sqrt{2} - 1)a &= t(A) - t(B) \\
&\leq \left| g\left(\text{sMoE}_f^\Phi(B)\right) - g\left(\text{sMoE}_f^\Phi(A)\right) \right| + \max\{1, t(B)/20\} + \max\{1, t(A)/20\} \\
&\leq L_g L_f \sqrt{m} \left\| D(B)^T B - D(A)^T A \right\|_2 + a/5.
\end{aligned}
$$

This implies that

$$a/10 \leq L_g L_f \sqrt{m} \left\| D(B)^T B - D(A)^T A \right\|_2 .$$

Taking the limit as $a \to \infty$ on either side of the above equation, and using Eq. equation 4, yields a contradiction.

We now consider the case where $\Phi_1 < 0$. The proof for this case is completely symmetric to the proof for the case of $\Phi_1 > 0$, and so it is omitted.

We now consider the final case of $\Phi_1 = 0$. Define $B \in \mathbb{R}^{m \times d}$ to be the matrix of all zeros, except $B_{11} = a$ for a value of $a > 0$ that is yet to be specified. Define $A \in \mathbb{R}^{m \times d}$ to be the matrix of all zeros, except $A_{11} = A_{21} = a/2$ for the aforementioned value of $a > 0$. Then since $A\Phi = B\Phi = 0$, we have

$$D(A) = D(B) = \begin{bmatrix} 1/m \\ 1/m \\ \vdots \\ 1/m \end{bmatrix}.$$

Let $A_1, B_1$ denote the first column of $A, B$. The above equality implies that

$$\left\| D(B)^T B - D(A)^T A \right\|_2 = \left| D(B)^T B_1 - D(A)^T A_1 \right| = \frac{a}{m} - \left( \frac{a}{2m} + \frac{a}{2m} \right) = 0.$$

Note via definition of $t$ that $t(B) - t(A) = (1 - 1/\sqrt{2})a$. Now recalling Eq. equation 2 and Eq. equation 3, we know for large $a > 0$ that

$$\begin{aligned}
(1 - 1/\sqrt{2})a &= t(B) - t(A) \\
&\leq g\left( \text{sMoE}_f^\Phi(B) \right) - g\left( \text{sMoE}_f^\Phi(A) \right) + \max\{1, t(B)/20\} + \max\{1, t(A)/20\} \\
&\leq L_g L_f \sqrt{m} \left\| D(B)^T B - D(A)^T A \right\|_2 + a/10 \\
&= a/10.
\end{aligned}$$

Thus we have arrived at a contradiction in each of the three cases. This completes the proof. ∎

## B CAN SOFT MOE WITH MULTIPLE EXPERTS REPRESENT $\| \cdot \|_2$?

Our Theorem 1 shows that Soft MoE with a single expert cannot represent the function $t$ defined as $t(X) = \|X\|_2$, even when the expert is arbitrarily powerful. However, it leaves open the possibility that Soft MoE with multiple experts could represent this function $t$. In this section, we provide a simple experiment to empirically check this. We consider the same problem of learning the function $t$, and we train a suite of Soft MoE models, each with a different number of experts, $n$, where $n \in \{1, 2, 5, 10\}$. As always (and as subsequently discussed), we fix the total expert parameter count.

We consider the input $X \in \mathbb{R}^{10}$, where $X \sim \mathcal{N}(0, 5I)$. For an input vector $X$, the label $y$ was set to $\|X\|_2$. Each batch of data is newly generated from this data distribution. We do two experiments with two different tokenization strategies. Concretely, the input was tokenized into either $X \in \mathbb{R}^{5 \times 2}$ or $X \in \mathbb{R}^{2 \times 5}$, i.e. 5 tokens each with dimension 2, or 2 tokens each with dimension 5, by partitioning the 10 elements of an input vector in order of their dimension index.

In the spirit of Theorem 1, the Soft MoE models considered consisted only of a Soft MoE layer, and a non-trainable summation prediction head. Therefore, given an input matrix $X \in \mathbb{R}^{m \times d}$ (i.e., $m$ tokens each with dimension $d$), it is passed directly to the Soft MoE layer. Each expert was a two layer MLP that had input and output of dimension $d$ and a single hidden layer with dimension $10d/n$. Thus, the number of expert parameters in each model was always constant at $20d^2$, regardless of the number of experts $n$ in the model. The output from the Soft MoE layer was then summed to produce the final scalar prediction from the model. Each model was trained with a batch size of $10,000$, with the Adam optimizer using default hyperparameters and learning rate of $1e\text{-}3$ over 500 epochs.

Figure 4 shows the loss curves from the 2 settings of token dimensions considered. We note that because each training batch is newly generated from the data distribution, the loss curves represent

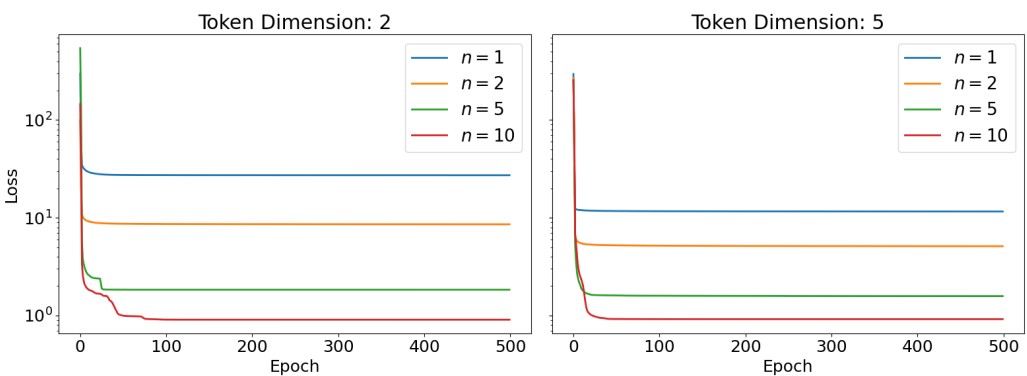

Figure 4: Loss curves in training Soft MoE models to learn the L2 norm function

both the train and test loss. The blue curve with $n = 1$ shows that having 1 large expert is unable to lower the loss and learn the L2 norm function, and thus provides empirical support for Theorem 1.

Notably, however, using more than one expert significantly reduces the loss, even though the number of expert parameters in each model was held constant. Indeed, we see in Figure 4 that increasing the number of experts seems to monotonically improve performance, even though the expert parameter count is fixed. This result therefore challenges the conventional wisdom that Soft MoE's empirical success with $n > 1$ smaller experts (each with $b/n$ parameters) is simply because they mimic the representation power of a single large expert (with $b$ parameters). Instead, it suggests that the Soft MoE has implicit biases, that assist in its improved performance.

## C   ADDITIONAL ALGORITHM DETAILS

We provide an example implementation of a batched version of Algorithm 1 where we assume we are given an input batch of size $b$. In describing each step, we will use `PyTorch`(Ansel et al., 2024) APIs, but we note that analogous functionalities to efficiently handle batch computation are readily available in other packages, such as `NumPy`(Harris et al., 2020), `TensorFlow`(Abadi et al., 2015), or `JAX`Bradbury et al. (2018).

---

**Algorithm 2** Best Expert Subset Selection for Batch of Inputs

---

**Require:** number of experts $k$ to use for prediction, Soft MoE $\text{sMoE}^{\Phi}_{\{f_j\}_{j=1}^n}$, input $X \in \mathbb{R}^{b \times m \times d}$

1: Compute batched $C(X) \in \mathbb{R}^{b \times m \times n}$.
    `C_X = torch.softmax(torch.matmul(X, Phi), dim=2)`
2: Compute batched $C_{\text{sum}} \in \mathbb{R}^{b \times n}$.
    `C_sum = torch.sum(C_X, dim=1)`
3: Compute batched $S_k \in [n]^{b \times k}$.
    `S_k = torch.argsort(C_sum, dim=1)[:, :k]`
4: Define a placeholder for batched $\widehat{Y}(X) \in \mathbb{R}^{b \times n \times d}$.
    `hat_Y_X = torch.zeros(b, n, d)`
5: For each expert $j \in [n]$, process the *subbatch* of $X$ that had selected expert $j$ according to $C_{\text{sum}}$.
    `for j in range(n):`
        `subbatch_idxs = (S_k == j).any(dim=1)`
        `subbatch = D_X[subbatch_idxs].transpose(1, 2) @`
    `X[subbatch_idxs]`
        `expert_output = f_i(subbatch)`
        `hat_Y_X[subbatch_idxs] = expert_output`
6: **return** $\widehat{Y}(X)$.

---

# D   DETAILS OF EMPIRICAL EXPERIMENTS

## D.1   DETAILS OF EXPERIMENTS IN SECTION 4.2

For this experiment on MNIST, the models used consisted of a single Soft MoE layer followed by a linear prediction head. Tables 4 and 5 provide a summary of the model and training procedure. To elaborate further, each model is trained with the Adam optimizer (Kingma & Ba, 2015) using default optimizer parameters for 15 epochs. We trained with the standard train set of $60,000$ datapoints and used the test set of $10,000$ datapoints for evaluation. Each setting of $n$ (number of experts) reached $\approx 97.5\%$ test accuracy after 15 epochs. During a forward pass of the model, each input image is tokenized into 4 patches, each of dimension $1 \times 14 \times 14$. Thus the input is $X \in \mathbb{R}^{4 \times 196}$, i.e. 4 tokens, each of dimension 196. For a network with $n$ experts, a single expert is an MLP with one hidden layer and ReLU activation, where the input and output are each 196 dimensional, and the number of hidden units is $196 \times 4/n$. While we trained a suite of 9 models with $n \in \{2^1, 2^2, 2^3, 2^4, 2^5, 2^6, 2^7, 2^8, 2^9\}$, the experiment in Section 4.2 uses 3 of these models with $n \in \{2^2, 2^3, 2^4\}$.

Note that the results in Table 1 show that for $n = 8$ and $n = 16$ the best $n/4$ expert subset actually predicts slightly better than just using the original network, i.e. utilizing all the experts. This is not too surprising, given that MNIST is relatively simple and our Soft MoE network is very overparameterized, and so exhaustively searching over many subnetworks can easily yield improved performance.

| Model Feature | Used Value |
|---|---|
| Raw Input Size | $1 \times 28 \times 28$ |
| Input Resizing | None |
| Soft MoE Input Size | $1 \times 28 \times 28$ |
| Patch Size | $1 \times 14 \times 14$ |
| Total Model Parameters | 318K |
| Total Expert Parameters | 307K |
| Number of Soft MoE Layers | 1 |
| Parameters per Expert | 307K $/n$ |
| Models Trained | $n \in \{2^1, 2^2, 2^3, 2^4, 2^5, 2^6, 2^7, 2^8, 2^9\}$ |

Table 4: Model details for MNIST experiments in Sections 4.2 and 4.4.

| Hyperparameter | Used Value |
|---|---|
| Batch Size | 256 |
| Epochs | 15 |
| Optimizer | Adam |
| LR | $1e$-3 |
| LR Scheduling | None: Constant LR |
| Epochs | 15 |

Table 5: Key hyperparameter settings for MNIST experiments in Sections 4.2 and 4.4.

In Figure 5, we display the number of unique subsets of the experts that were used to produce the best $k$-subset accuracies in Table 1. Since there are $10,000$ datapoints in the test set and $\binom{n}{k}$ total number of unique subsets, each bar is capped at $\min\{10,000, \binom{n}{k}\}$.

As there are few combinations of expert subsets of size $k = n/4$ with low $n$, all subset combinations are required to produce the best accuracies when $n$ is low, but they still do not achieve perfect accuracies, as indicated in the first row of Table 1. The number of unique subsets grows with $n$, which indicates an increase in the diversity of experts used per datapoint, and thus more degree of specialization of the experts to each input $X$ (following our notion of specialization from Section 4.1).

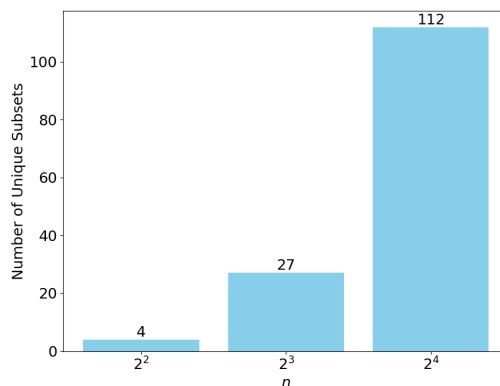

Figure 5: Each bar presents the number of unique subsets of experts of size $k = n/4$ that were used to produce the highest accuracy, for each setting of the total number of experts $n$.

## D.2 DETAILS OF EXPERIMENTS IN SECTION 4.4

The MNIST experiment in this section re-used the models that were trained from Section 4.2. In this set of experiments, we used all 9 models, each with a different number of experts.

For the CIFAR10 experiment, we trained a suite of Soft MoE models with different numbers of experts $n$ by taking the Astroformer-1 model as backbone and replacing the MLPs of the transformer blocks with a Soft MoE layer. The Astroformer model assumes the "C-C-C-T" architecture, where "C" stands for convolution and "T" stands for transformer. The transformer stage consists of 2 transformer blocks, each of which has a feedforward network. We replace this feedforward network with a Soft MoE layer. An input image of size $3 \times 96 \times 96$ (channel $\times$ height $\times$ width) is processed by the "C" stages into a tensor of size $768 \times 3 \times 3$ when it reaches the first "T" stage. Following standard practice in tokenizing images via patches, we treat each height-width location as a token, and thus create 9 total tokens (or patches), each with dimension 768. Each expert is a 2-layer MLP with inputs and outputs that are both 768 dimensional, and the number of hidden units is $768 * 64/n$. A summary of the model features are provided in Table 6.

| Model Feature | Used Value |
|---|---|
| Raw Input Size | $3 \times 32 \times 32$ |
| Resized Input Size | $3 \times 96 \times 96$ |
| Soft MoE Input Size | $768 \times 3 \times 3$ |
| Patch Size | $768 \times 1 \times 1$ |
| Base Model Architecture | Astroformer-1 |
| Total Model Parameters | 180M |
| Total Expert Parameters | 150M |
| Number of Soft MoE Layers | 2 |
| Parameters per Expert | 150M/$2n$ |
| Models Trained | CIFAR10: $n \in \{2^1, 2^2, 2^3, 2^4, 2^5, 2^6, 2^7\}$ CIFAR100: $n \in \{2^1, 2^3, 2^4, 2^6, 2^7\}$ |

Table 6: Model details for CIFAR10 and CIFAR100 experiments in Section 4.4.

One additional modification we had to make to this architecture was that there are no residual connections from directly before Soft MoE to directly after Soft MoE. All of the other residuals connections between other blocks were retained. We made this design choice because we found that our Soft MoE variant of Astroformer trains in a manner that essentially ignores expert outputs and only relies on the residual connections (i.e., the expert outputs are negligible compared to the residual portion). Such a scenario biases the results, since the remainder of the network makes predictions without relying on the Soft MoE layer at all, and we thus cannot assess Algorithm 1. While this may

| Hyperparameter | Used Value |
|---|---|
| Batch Size | 800 |
| Gradient Accumulation Steps | 1 |
| Epochs | 500 |
| Optimizer | AdamW |
| LR Scheduler | Cosine |
| Base LR | $3e\text{-}4$ |
| LR Cycle Decay | $1e\text{-}2$ |
| LR K Decay | 1 |
| Warmup LR | $1e\text{-}5$ |
| Epochs | 500 |
| Warmup Epochs | 5 |
| Mixup | 0.8 |
| Label Smoothing | 0.1 |
| Dropout Rate | 0.1 |

Table 7: Key hyperparameters used for CIFAR10 and CIFAR100 experiments in Section 4.4.

be an artifact of hyperparameter settings that are not optimized for our Soft MoE variant models, we did not have the compute surplus to perform an exhaustive search over hyperparameters and utilized those that were provided with the model implementation and code.

To train our Soft MoE variant of the Astroformer-1 models, we followed the exact same training procedure as that provided in the Astroformer paper (Dagli, 2023) and their public codebase[1], with the exception of the input resizing and base learning rate. While there are many hyperparameter settings that were set by default when we used their codebase and commands, we provide a summary of a few key settings in Table 7. We trained with the standard train set of $50,000$ datapoints and used the test set of $10,000$ datapoints for evaluation. Training was halted as soon as the model crossed $95\%$ test accuracy, which was typically well before the total number of epochs.

For the CIFAR100 experiment, we used the exact same model and training code as that of CIFAR100. In our trials of training the base Astroformer-1 model (i.e. without any MoE layers or modifications), the model converges to $\approx 80\%$ test accuracy over 500 epochs of training, and our Soft MoE variants of Astroformer-1 also converged to $\approx 80\%$ test accuracy using the same training procedure and code.

For the ImageNet-1k experiment, we used the same model architecture as that used by the seminal paper that proposed Soft MoE (Puigcerver et al., 2024). Specifically, we used the Soft MoE adaptation of the ViT Base model, with the only difference being that the expert MLPs are of different sizes, depending on the number of experts, $n$. We defer exact details of the architecture by referring the reader to the Soft MoE paper (Puigcerver et al., 2024) and previous works on ViTs (Dosovitskiy et al., 2021), but we re-iterate a few key features of the architecture here in text and in Table 8. Our model is based on the ViT Base model with patch size $16 \times 16$. This model consists of 12 encoder blocks in total, where each encoder block consists of an attention layer followed by an MLP. The MLP of the last 6 out of 12 encoder blocks have been replaced with a Soft MoE layer, where each expert is a 2-layer MLP. The input and output of each expert MLP is 768-dimensional, and there is one hidden layer of dimension $768 * 64/n$. In implementing these models, we relied on a publicly available PyTorch implementation of Soft MoE variant of ViT[2].

Since there are no publicly available pretrained weights for the Soft MoE variant of ViT models, we had to train each model from scratch. We relied on the DeiT (Data-efficient Image Transformers) (Touvron et al., 2021; 2022) training procedure and code, since our compute constraints made pretraining on a large corpus of data infeasible. Specifically, we used the "'DeiT-III'" training procedure, which is publicly available [3]. We used the same command that was used to train the "deit_base_patch16_LS" model on ImageNet-1k with just one change: we trained without resizing the inputs from 224 to 192. A few key hyperparameter settings used are listed in Table 9.

---

[1] https://github.com/Rishit-dagli/Astroformer

[2] https://github.com/bwconrad/soft-moe

[3] https://github.com/facebookresearch/deit/blob/main/README_revenge.md

| Model Feature | Used Value |
|---|---|
| Raw Input Size | $3 \times 224 \times 224$ |
| Input Resizing | None |
| Patch Size | $3 \times 16 \times 16$ |
| Soft MoE Input Size | $197 \times 768$ |
| Base Model Architecture | ViT Base |
| Total Model Parameters | 513M |
| Total Expert Parameters | 454M |
| Number of Soft MoE Layers | 6 |
| Parameters per Expert | 454M/$6n$ |
| Models Trained | $n \in \{2^1, 2^3, 2^4, 2^6, 2^7\}$ |

Table 8: Model details for ImageNet-1k experiments in Section 4.4.

| Hyperparameter | Used Value |
|---|---|
| Batch Size | 512 |
| Gradient Accumulation Steps | 1 |
| Epochs | 800 |
| Optimizer | FusedLAMB |
| LR Scheduler | Cosine |
| Base LR | $3e$-3 |
| Warmup LR | $1e$-6 |
| Epochs | 500 |
| Warmup Epochs | 5 |
| Random Erase Prob | 0.0 |
| Mixup | 0.8 |
| Cutmix | 1.0 |
| Label Smoothing | 0.0 |
| Dropout Rate | 0.0 |
| Weight Decay | 0.05 |

Table 9: Key hyperparameters settings for ImageNet-1k experiments in Section 4.4.

We trained with the standard train set of 1.3M datapoints and used the validation set of $50,000$ datapoints for evaluation. The DeiT-III code and provided command can train the original ViT Base model (i.e. without any MoE layers or modifications) to $\approx 81\%$ validation accuracy over 800 training epochs. Our Soft MoE ViT models are able to reach $\approx 79\%$ validation accuracy within 800 epochs. We note that this code and the hyperparameters set are highly optimized for training original ViT models, thus it is not surprising that our models are not able to reach or exceed the validation accuracy of the original ViT model.

We had access to 8 NVIDIA A6000 GPUs to train all of our models.

## D.3    ADDITIONAL RESULTS FOR SECTION 4.4.2

This section provides additional results for the experiments discussed in Section 4.4.2. Specifically, we provide two additional sets of tables for each of the datasets: 1) the number of standard deviations by which the test accuracy of Algorithm 1 exceeds the mean test accuracy of random expert subset selection, and 2) the test accuracy of Algorithm 1 as a percentage of the original accuracy when using all of the experts without any expert dropping. While Section 4.4.2 provides partial results for CIFAR100, we repeat the result here for completeness, and provide the full set of results for MNIST (Table 10), CIFAR10 (Table 11), CIFAR100 (Table 12), and ImageNet-1k (Table 13).

| | $2^1$ | $2^2$ | $2^3$ | $2^4$ | $2^5$ | $2^6$ | $2^7$ | $2^8$ | $2^9$ |
|---|---|---|---|---|---|---|---|---|---|
| | | | | | $n$ | | | | |
| $k = n/2$ | 25.79 | 33.61 | 22.95 | 25.39 | 23.33 | 26.47 | 22.98 | 10.19 | 18.70 |
| $k = n/4$ | - | 0.00 | 16.33 | 45.83 | 38.89 | 49.00 | 37.15 | 31.33 | 29.44 |
| $k = n/8$ | - | - | 11.81 | 28.24 | 46.78 | 38.78 | 50.46 | 42.34 | 66.90 |

(a) Number of standard deviations that the performance of Algorithm 1 is above the mean performance of Random expert set selection. For the Random selection, we used 10 random seeds to obtain its mean and standard deviation.

| | $2^1$ | $2^2$ | $2^3$ | $2^4$ | $2^5$ | $2^6$ | $2^7$ | $2^8$ | $2^9$ |
|---|---|---|---|---|---|---|---|---|---|
| | | | | | $n$ | | | | |
| $k = n/2$ | 78.1% | 80.3% | 88.5% | 94.2% | 96.6% | 97.0% | 97.3% | 98.2% | 99.1% |
| $k = n/4$ | - | 47.7% | 60.8% | 76.2% | 86.4% | 88.4% | 90.2% | 93.2% | 96.0% |
| $k = n/8$ | - | - | 40.3% | 51.9% | 71.6% | 76.2% | 80.4% | 85.9% | 90.6% |

(b) Test accuracy of Algorithm 1 as a percentage of the test accuracy of using all experts.

Table 10: Additional results for MNIST experiments.

| | $2^1$ | $2^2$ | $2^3$ | $2^4$ | $2^5$ | $2^6$ | $2^7$ |
|---|---|---|---|---|---|---|---|
| | | | | $n$ | | | |
| $k = 5$ | - | - | 1.73 | 0.95 | 3.88 | 7.25 | 25.90 |
| $k = 4$ | - | - | 1.99 | 2.51 | 4.74 | 15.12 | 46.92 |
| $k = 3$ | - | 0.53 | 3.47 | 7.81 | 12.28 | 21.53 | 88.96 |
| $k = 2$ | - | 4.05 | 12.26 | 15.17 | 24.26 | 42.77 | 176.57 |
| $k = 1$ | 3.22 | 20.40 | 33.14 | 35.48 | 70.75 | 83.71 | 234.04 |

(a) Number of standard deviations that the performance of Algorithm 1 is above the mean performance of Random expert set selection. For the Random selection, we used 10 random seeds to obtain its mean and standard deviation.

| | $2^1$ | $2^2$ | $2^3$ | $2^4$ | $2^5$ | $2^6$ | $2^7$ |
|---|---|---|---|---|---|---|---|
| | | | | $n$ | | | |
| $k = 5$ | - | - | 100% | 99.9% | 100% | 100% | 99.9% |
| $k = 4$ | - | - | 100% | 100% | 99.9% | 99.9% | 99.9% |
| $k = 3$ | - | 99.9% | 99.9% | 99.9% | 99.9% | 100% | 99.8% |
| $k = 2$ | - | 99.9% | 99.9% | 99.9% | 99.8% | 99.9% | 99.7% |
| $k = 1$ | 100% | 99.9% | 99.7% | 99.8% | 99.9% | 99.5% | 96.0% |

(b) Test accuracy of Algorithm 1 as a percentage of the test accuracy of using all experts.

Table 11: Additional results for CIFAR10 experiments.

| | | $n$ | | | |
|---|---|---|---|---|---|
| | $2^1$ | $2^3$ | $2^4$ | $2^6$ | $2^7$ |
| $k = n/2$ | 6.23 | 6.73 | 1.89 | 3.47 | 2.49 |
| $k = n/4$ | - | 7.31 | 6.34 | 4.56 | 4.50 |
| $k = n/8$ | - | 21.22 | 28.20 | 13.96 | 5.74 |

(a) Number of standard deviations that the performance of Algorithm 1 is above the mean performance of Random expert set selection. For the Random selection, we used 10 random seeds to obtain its mean and standard deviation.

| | | $n$ | | | |
|---|---|---|---|---|---|
| | $2^1$ | $2^3$ | $2^4$ | $2^6$ | $2^7$ |
| $k = n/2$ | 99.6% | 99.9% | 99.8% | 99.8% | 99.8% |
| $k = n/4$ | - | 95.8% | 98.9% | 99.1% | 99.2% |
| $k = n/8$ | - | 85.0% | 94.2% | 94.4% | 95.6% |

(b) Test accuracy of Algorithm 1 as a percentage of the test accuracy of using all experts.

Table 12: Additional results for CIFAR100 experiments.

| | | $n$ | | | |
|---|---|---|---|---|---|
| | $2^1$ | $2^3$ | $2^4$ | $2^6$ | $2^7$ |
| $k = n/2$ | 2.23 | 6.72 | 1.63 | 24.54 | 35.50 |
| $k = n/4$ | - | 20.71 | 14.06 | 38.89 | 32.11 |
| $k = n/8$ | - | 31.88 | 20.21 | 35.04 | 60.38 |

(a) Number of standard deviations that the performance of Algorithm 1 is above the mean performance of Random expert set selection. For the Random selection, we used 10 random seeds to obtain its mean and standard deviation.

| | | $n$ | | | |
|---|---|---|---|---|---|
| | $2^1$ | $2^3$ | $2^4$ | $2^6$ | $2^7$ |
| $k = n/2$ | 98.6% | 99.2% | 99.1% | 99.9% | 99.9% |
| $k = n/4$ | - | 99.1% | 98.7% | 99.8% | 99.8% |
| $k = n/8$ | - | 98.9% | 98.2% | 99.5% | 99.8% |

(b) Test accuracy of Algorithm 1 as a percentage of the test accuracy of using all experts.

Table 13: Additional results for ImageNet-1k experiments.

## D.4    ADDITIONAL ANALYSIS FOR SECTION 4.4.2

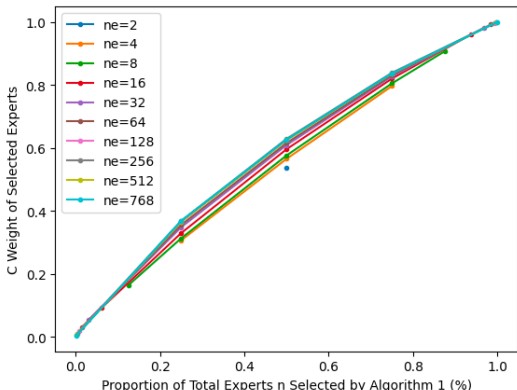

Figure 6: Sum of the $C_{\text{sum}}$ weight allocated to the highest weighted experts tends to increase with the total number of experts $n$. However, the increase in the allocated weight is not significant, even across very different values of $n$. For example, when $n = 8$, the top 50% of the highest weighted experts by $C_{\text{sum}}$ were allocated about 0.55 weight on average. That figure increased to about 0.57 for $n = 128$, which indicates that the naive argument that the effectiveness of Algorithm 1 is simply due to dropping marginally weighted expert outputs is unlikely.

In Section 4.4.2, we showed that $C_{\text{sum}}$ (which is derived from $C(X)$) becomes more informative in identifying the specialized subset of experts as the total number of experts $n$ increases. If the skewness of $C_{\text{sum}}$ increased with $n$ such that the weights of $C_{\text{sum}}$ became highly concentrated on a select few experts with large $n$, this would provide a trivial explanation for the performance of Algorithm 1: increasingly marginally weighted experts are dropped. We show that this is not the case.

In Figure 6, we provide a visualization of the sum of the weight of $C_{\text{sum}}$ allocated to the highest weighted experts. This figure was produced by using the same set of models that were used to produce Figures 1 and 2a, and the sum of the $C_{\text{sum}}$ weights were averaged across the test dataset. While the $C$ weight tends to increase with $n$, the increase is marginal, even across very different values of $n$. The minor differences across the drastically different values of $n$ indicate that the specialization identified via Algorithm 1 is not trivial, and that the specialization identified is not merely through each experts' linear weight in the contribution to the output.

## D.5    DETAILS OF EXPERIMENTS IN SECTION 4.5

For this experiment, we consider the same Soft MoE variant of the ViT Base architecture as those used in Section 4.4, but with larger experts to make the total parameter count larger. Specifically, each of the Soft MoE layers has 8 experts, each of which are 2 layer MLP with input and output dimensions of 768 and a hidden layer with $768 * 40$ units. The whole model has 2.32B parameters, of which 2.27B parameters are in the experts.

The data used for the forward pass was generated from a standard normal distribution. The experiment was done on a single NVIDIA GeForce RTX 2080 Ti GPU. Latency was measured by first performing 100 warmup forward passes, then synchronizing CUDA, then measuring the elapsed wall clock time during the next 100 forward passes plus the end of another synchronization of CUDA. The reported figures in Table 3 are based on these latter 100 forward passes. While this experiment is simple and relatively small-scale, a proper analysis would require industry-scale models, and would be dependent on the hardware and the extent of distributed computing and per-device parallelization, as discussed in Section 4.5. We lack the compute capacity to perform such a thorough study.

