# OpenReview forum: "Beyond Parameter Count: Implicit Bias in Soft Mixture of Experts"
_ICLR.cc/2025/Conference — Submitted to ICLR 2025_

### Official Review · Reviewer_PNpP · 2024-10-27

**Soundness:** 3
**Presentation:** 3
**Contribution:** 2
**Rating:** 5
**Confidence:** 4

**Summary:**

In this paper, the authors investigate the representation power of the soft Mixture-of-Experts (MoE) model when using a single large expert versus when using multiple smaller experts, and then explore the expert specialization in this model. In particular, they theoretically demonstrate that the soft mixture of a single expert cannot represent simple target functions, while they empirically show that the soft mixture of multiple smaller experts with the same parameter count is capable of this task. Thus, they claim that even for a fixed parameter count, using multiple experts is necessary for achieving a good representation power rather than just help reduce the computational overheads as in the principle of traditional sparse MoE. Subsequently, they introduce the definition of expert specialization for the soft MoE and propose an algorithm to choose the best subset of experts (including a quarter of the original number of experts) specializing in predicting the label of an input.

**Strengths:**

1. The comparison between using a single large expert versus using multiple smaller experts is of interest and provide a useful insight into designing the optimal MoE structure. Additionally, the algorithm for selecting the subset of specialized experts in an input potentially helps reduce the computation burden.

2. The presentation of this paper is good (except for the statement of Theorem 1), which makes it easy to follow.

3. The theoretical proofs and experiments are conducted in a careful way.

**Weaknesses:**

1. The result in Theorem 1 is not really solid as the upper bound still hinges upon the input value. Moreover, since the authors choose a specific target function rather than consider an arbitrary convex function, I am quite skeptical about the generalization of the result. Additionally, the way this theorem is stated confuse me when I first read it. I think the writing of the statement could be improved.

2. The paper lacks the theoretical proof for the result that the soft mixture of multiple small experts is capable of representing simple target functions, which weakens the contribution of this paper considerably. Personally, the experiments on the synthetic data in Appendix B for empirically showing this result is not sufficient to convince me.

3. The principle of the proposed algorithm for choosing the best subset of important experts based on their weights in the matrix $C(X)$ seems to be reasonable. However, I believe that the claim that this algorithm can help reduce computation at inference would be more convincing if the experiments are conducted on bigger model and dataset.

**Questions:**

1. What are the challenges of generalizing the result in Theorem 1 to the setting when the target function is assumed to belong to some class of convex functions rather than selected specifically?

2. What are the challenges of theoreticall show that the soft mixture of multiple small experts with a fixed parameter count can represent a simple target function?

---

### Official Review · Reviewer_zVi5 · 2024-11-03

**Soundness:** 3
**Presentation:** 4
**Contribution:** 3
**Rating:** 6
**Confidence:** 4

**Summary:**

This paper explores the Soft Mixture of Experts (Soft MoE) architecture, concentrating on its inherent biases and constraints in representational power. Through rigorous theoretical and empirical analyses, the authors show that a single expert within the Soft MoE framework is insufficient to represent even basic convex functions, highlighting the need for multiple experts to achieve optimal performance. The paper presents Theorem 1, which formally substantiates these limitations, and supports it with synthetic data experiments, illustrating that increasing the number of experts enhances performance. Additionally, the authors introduce an efficient method (Algorithm 1: Best Expert Subset Selection) for selecting specialized expert subsets, designed to reduce computational demands in large-scale applications while maintaining model efficacy.

**Strengths:**

- The originality lies in the paper’s creative approach, combining established concepts in expert modeling with a critical examination of representational limitations, as demonstrated by Theorem 1 and the empirical evidence in Appendix B.

- This work has practical significance for advancing the design of scalable MoE models. By addressing the limitations in single-expert representational capacity, the paper opens up discussions on how architectural adjustments, such as multiple experts or efficient selection of subsets, can improve performance and reduce computational costs. These insights are valuable for both researchers and practitioners interested in efficient, scalable solutions for high-dimensional applications in MoE architectures.

**Weaknesses:**

- **Clarity in Methodological Exposition:** Certain aspects of the methodology, such as the approach for selecting the number of experts $k$ for prediction in real applications, would benefit from more detailed and clearer explanations.

- **Clarity in Technical Proof Exposition:** Including a proof sketch at the beginning of Appendix A would enhance readability and provide a clearer roadmap for understanding the technical details. Additionally, a more detailed explanation of why the equalities in lines 711-716 hold would further improve clarity and support the reader's comprehension of the argument. Clarification is needed regarding the notation for norms: are $|\cdot|$ and $\| \cdot \|_2$ intended to represent the same norm?

- **Limited Theoretical Analysis for Multi-Expert Configurations:** Although the paper offers theoretical insights into the limitations of a single expert, it does not extend the theoretical framework to cover multiple-expert $n$ or multiple-slot configurations. A more detailed analysis of how increasing the number of experts impacts scalability, convergence, or approximation accuracy would enhance the strength of the claims presented.

- **Notation Section**: The manuscript would benefit from a dedicated notation section to clarify certain symbols that may be unclear, such as the norm $|\cdot|$ or $\| \cdot \|_2$ for matrices and vectors. Providing a notation section would enhance readability and ensure that readers can follow the mathematical expressions consistently throughout the paper.

- **More Accurate References:** **Lines 040 and 549-550**: The correct citation should be: "Yasaman Bahri, Ethan Dyer, Jared Kaplan, Jaehoon Lee, and Utkarsh Sharma (2024). Explaining neural scaling laws. Proceedings of the National Academy of Sciences, 121(27), e2311878121." **Lines 073 and 633-635**: The correct citation should be: "Johan Obando-Ceron, Ghada Sokar, Timon Willi, Clare Lyle, Jesse Farebrother, Jakob Foerster, Gintare Karolina Dziugaite, Doina Precup, and Pablo Samuel Castro. Mixtures of experts unlock parameter scaling for deep RL, 2024. Proceedings of the 41st International Conference on Machine Learning, PMLR 235:38520-38540, 2024." **Lines 073 and 571-574**: The correct citation should be: "Damai Dai, Chengqi Deng, Chenggang Zhao, R. X. Xu, Huazuo Gao, Deli Chen, Jiashi Li, Wangding Zeng, Xingkai Yu, Y.Wu, Zhenda Xie, Y. K. Li, Panpan Huang, Fuli Luo, Chong Ruan, Zhifang Sui, and Wenfeng Liang. Deepseekmoe: Towards ultimate expert specialization in mixture-of-experts language models, 2024. In Proceedings of the 62nd Annual Meeting of the Association for Computational Linguistics (Volume 1: Long Papers), pages 1280–1297, Bangkok, Thailand. Association for Computational Linguistics."

**Questions:**

- **Extension to Multiple Slots per Expert**: From a technical standpoint, are there any challenges preventing the authors from considering only a single slot per expert? If multiple slots $p$, where $p \ge 2$, are considered, would Theorem 1 and the related results still hold? The same question applies to the benign target $t(X) = ||X||_2$: does the result hold only for this specific function, or can it be extended to more general target functions?

- **Multiple Experts Results**: I believe the experiment in Appendix B can be explained by leveraging the approximation properties of functions through mixture of experts models as they relate to the number of experts, as discussed in [1, 2, 3, 4, 5]. At a minimum, the authors should provide a more detailed discussion and/or comparisons with the mentioned references to offer greater insights into how using multiple experts significantly reduces the loss, even when the number of expert parameters in each model is kept constant.

References:

[1] Mendes, E. F., \& Jiang, W. (2012). On convergence rates of mixtures of polynomial experts. Neural computation, 24(11), 3025-3051.

[2] Jiang, W., \& Tanner, M. A. (1999, January). Hierarchical mixtures-of-experts for generalized linear models: some results on denseness and consistency. In Seventh International Workshop on Artificial Intelligence and Statistics. PMLR.

[3] Nguyen, H. D., Lloyd-Jones, L. R., \& McLachlan, G. J. (2016). A universal approximation theorem for mixture-of-experts models. Neural computation, 28(12), 2585-2593.

[4] Nguyen, H. D., Nguyen, T., Chamroukhi, F.,\& McLachlan, G. J. (2021). Approximations of conditional probability density functions in Lebesgue spaces via mixture of experts models. Journal of Statistical Distributions and Applications, 8, 1-15.

[5] Nguyen, H. D., Chamroukhi, F., \& Forbes, F. (2019). Approximation results regarding the multiple-output Gaussian gated mixture of linear experts model. Neurocomputing, 366, 208-214.

---

### Official Review · Reviewer_kufN · 2024-11-04

**Soundness:** 2
**Presentation:** 2
**Contribution:** 2
**Rating:** 3
**Confidence:** 4

**Summary:**

This paper explores implicit biases in Soft Mixture of Experts (Soft MoE) models, challenging traditional assumptions about their representation power and specialization. Unlike prior MoE work focused on computational efficiency, this paper investigates whether the Soft MoE’s continuous gating mechanism disrupts specialization. In particular, the main contributions are as follows:

- **Single Expert Limitation:** Proving that a Soft MoE with one large expert cannot approximate $t(X) = ||X||_2$, claiming that multiple experts are essential for effective representation.

- **Expert Specialization:** Demonstrating that adding more experts introduces an implicit bias toward specialized subsets for specific inputs.

- **Efficient Inference:** Proposing a method to select specialized experts during inference.

**Strengths:**

The paper considers important and relevant questions regarding implicit biases in Soft MoEs, particularly regarding expert specialization. I find these questions relevant and interesting for the ML community.

**Weaknesses:**

- **W1. SoftMoE Formulation**: The SoftMoE formulation presented in Section 2 of this paper differs from that of the original SoftMoE paper [1]. In the original formulation, each expert processes $p$ slots, making $ \Phi \in \mathbb{R}^{d \times (n \cdot p)}$ and $C(X) \in \mathbb{R}^{m \times (n \cdot p)}$. However, in this paper, the parameter $p$ is not included. Notably, setting $p=1$ in the original formulation results in an incomparability with other papers using SoftMoE, where $p$ is recognized as a significant hyperparameter. Furthermore, in line 243, the authors cite [2], claiming that this work demonstrates that even a single-expert SoftMoE can outperform traditional architectures. This comparison is inaccurate, as $p$ is not set to 1. Specifically, after reviewing the code, I observed that $p \approx m/n$; see lines 55-64 in this GitHub file: [Link](https://github.com/google/dopamine/blob/master/dopamine/labs/moes/architectures/softmoe.py).

- **W2. Theoretical Contribution**: I don't find the results presented in Theorem 1 insightful.

    - **W2.1. Trivial Result**: Ignoring $p$ (the number of slots) in the SoftMoE formulation leads to a trivial outcome for $n=1$. Here, matrix $C(X)$ becomes an **all-one vector of size $m$** instead of a matrix of size $m \times p$, resulting in **identical output tokens** and thereby ignoring any temporal information in the tokens, effectively replacing them with a fixed token. This trivializes the ineffectiveness of SoftMoE with $n=1$, which is not the case when $p > 1$.

    - **W2.2. Notion of Approximation**: The notion of approximation presented by the authors is non-standard. I encourage the authors to provide references or pointers to similar approaches in the literature to clarify this aspect.

    - **W2.3. Proof Technique**: After reviewing Appendix A, I noticed that the proof relies on a special case where a contradiction arises as matrix norms approach infinity. This is acknowledged by the authors in Section 3, where they mention that normalizing the input makes the results from Theorem 1 inapplicable.

- **W3. Lack of comparison for Algorithm 1:** The paper lacks a comparison with the sparse routing mechanism considered in the original SoftMoE paper [1] (cf. Section 3.2). (Correct me if I am wrong)

---

References:
[1] Puigcerver, Joan, et al. "From sparse to soft mixtures of experts." arXiv preprint arXiv:2308.00951 (2023).
[2] Obando-Ceron, Johan, et al. "Mixtures of experts unlock parameter scaling for deep RL." arXiv preprint arXiv:2402.08609 (2024).

**Questions:**

- **Q1.** How does Algorithm 1 compare to the Token Choice and Expert Choice methods mentioned in the original SoftMoE paper?

---

### Official Review · Reviewer_LJZk · 2024-11-08

**Soundness:** 1
**Presentation:** 2
**Contribution:** 1
**Rating:** 3
**Confidence:** 4

**Summary:**

The paper theoretically analyses a recently introduced model known as soft Mixture of experts. It makes the claim that multiple experts are not just a tool for improving computational performance and perform a crucial role in shaping the representation power of the model architecture. The paper also introduces a notion of expert specialization for this architecture. It claims that empirically, based on this notion, most data points are taken care of by specialized experts, i.e. only a small fraction of the experts (dependent on the input) are sufficient to correctly classify the input.

**Strengths:**

The paper gives a good summary of the soft MoE model and sets up notation in a clear way. The notion of specialization is interesting. The proposed algorithm for selecting experts is simple and intuitive.

**Weaknesses:**

Section 3 on the representation failure of a single expert is quite trivial and obvious. Each expert essentially sees X only through a single d-dimensional linear projection, and hence any function of X that uses all of X in a non-linear manner clearly cannot be represented by a single expert regardless of how complex the expert is. Mutiple experts can potentially overcome this because they would each look at different projections.

Section 4 results on specialization are also quite weak. Choosing experts for each input separately can lead to an obviously absurd conclusion as follows. Having 10 experts each of whom predict a single class constantly would be considered highly specialised.

The proposed algorithm 1 is literally just picking the experts with the highest weight for each example, and is the first common sense thing that a practitioner looking to reduce inference complexity would do. The novelty of this approach is very minimal.

**Questions:**

Traditional analyses for specialisation/interpretability typically study the sparsity of the routing vector or the mass given to the top fraction (say) 10 percent,  over the duration of training. Would such an analysis be relevant here as well?

---

### Meta-Review · Area_Chair_sF46 · 2024-12-21

**Metareview:**

In the paper, the authors provide a theoretical analysis of the soft Mixture of Experts (MoE) model. They demonstrate that the usage of multiple experts not only enhances computational performance but also fundamentally shapes the representational power of the model. Furthermore, the authors also prove empirically that the experts are well-specialized.

After the rebuttal, there have remained several weaknesses of the current paper: (1) Some of the theoretical results are obvious, weak, and trivial (e.g., the results in Section 3 or Section 4, etc.). (2) The proposed algorithm lacks novelty and is not convincing. (3) The writing of the paper requires major revision.

Given the above weaknesses of the paper, I recommend rejecting the paper at the current stage. The authors are encouraged to incorporate the suggestions and feedback of the reviewers into the revision of their manuscript.

**Additional Comments On Reviewer Discussion:**

Please refer to the meta-review.

---

### Decision · Program_Chairs · 2025-01-22

Reject